# Prognostic Factors and the Role of Adjuvant Chemotherapy in Pathological Node-Negative T3 Gastric Cancer

**DOI:** 10.3390/jpm13030553

**Published:** 2023-03-20

**Authors:** Yi-Fu Chen, Ming-Yang Chen, Puo-Hsien Le, Tsung-Hsing Chen, Chia-Jung Kuo, Shang-Yu Wang, Shih-Chiang Huang, Wen-Chi Chou, Ta-Sen Yeh, Jun-Te Hsu

**Affiliations:** 1Department of General Surgery, Chang Gung Memorial Hospital at Linkou, College of Medicine, Chang Gung University, Taoyuan 33305, Taiwan; yifu061990@cgmh.org.tw (Y.-F.C.); hitsuzi@cgmh.org.tw (M.-Y.C.); m7026@cgmh.org.tw (S.-Y.W.); tsy471027@cgmh.org.tw (T.-S.Y.); 2Department of Gastroenterology, Chang Gung Memorial Hospital at Linkou, College of Medicine, Chang Gung University, Taoyuan 33305, Taiwan; b9005031@cgmh.org.tw (P.-H.L.); q122583@cgmh.org.tw (T.-H.C.); m7011@cgmh.org.tw (C.-J.K.); 3Department of Pathology, Chang Gung Memorial Hospital at Linkou, College of Medicine, Chang Gung University, Taoyuan 33305, Taiwan; ab8640112@cgmh.org.tw; 4Department of Hematology-Oncology, Chang Gung Memorial Hospital at Linkou, College of Medicine, Chang Gung University, Taoyuan 33305, Taiwan; f12986@cgmh.org.tw

**Keywords:** prognostic factor, gastric cancer, adjuvant chemotherapy, node-negative

## Abstract

The role of adjuvant chemotherapy in pathological T3N0M0 (pT3N0M0) gastric cancer (GC) remains unclear. The aim of this study was to analyze the prognostic factors of patients with pT3N0M0 GC and to clarify which ones could benefit from adjuvant chemotherapy. A total of 137 patients with pT3N0M0 GC were recruited between 1994 and 2020. Clinicopathological factors and adjuvant chemotherapy regimens were retrospectively collected. Prognostic factors of disease-free survival (DFS) and cancer-specific survival (CSS) were determined using univariate and multivariate analyses. The chemotherapy group was younger (*p* = 0.012), had had more lymph nodes retrieved (*p* = 0.042) and had higher percentages of vascular invasion (*p* = 0.021) or perineural invasion (*p* = 0.030) than the non-chemotherapy group. There were no significant differences in DFS (*p* = 0.222) and CSS (*p* = 0.126) between patients treated with or without adjuvant chemotherapy. Stump cancer, tumor size and perineural invasion were associated with higher rates of recurrence. Tumor size was an independent prognostic factor for DFS (hazard ratio, 4.55; confidence interval, 1.59–12.99; *p* = 0.005) and CSS (hazard ratio, 3.97; confidence interval, 1.38–11.43; *p* = 0.011). Tumor size independently influenced survival outcomes in pT3N0M0 patients who underwent radical surgery with and without adjuvant chemotherapy.

## 1. Introduction

Gastric cancer (GC) ranks as the fifth most common malignancy, with over one million newly diagnosed cases annually, and has the fourth highest cancer-related deaths, at over 700,000 persons annually worldwide [1]. Radical resection is still the current standard care for localized GC. The use of adjuvant chemotherapy is recommended for patients with Stage II-III disease to reduce recurrence and prolong overall survival [2,3]. However, the additional benefit from adjuvant chemotherapy for pathological T3N0M0 (pT3N0M0) patients is very limited, since those patients have over 80% five-year survival rates after radical surgery alone [4,5]. Furthermore, the TNM staging system is grouped according to five-year overall survival, not reflecting the tumor biological perspective. Therefore, researchers have suggested that clinicopathological factors other than TNM classification should be considered to evaluate the recurrence risk and survival [6], and adjuvant chemotherapy may be accordingly omitted in patients with low risk of relapse to decrease the unnecessary exposure to cytotoxic agents. Currently, for pT3N0M0 patients, the definition of “high risk” is not well defined, since some patients who have high-risk recurrence factors do not actually experience recurrence, whereas some with diseases that are deemed low risk do.

Few precise predicting tools in the assessment of recurrence risk in pT3N0M0 address this clinical dilemma, limiting treatment plans to the subgroup of patients with high-risk features, to whom they are most likely to confer a survival benefit. To date, many efforts to refine recurrence risks, except for clinicopathological factors, in localized GC have focused on examining surgical specimens with various biomarkers. Although such tissue-based biomarkers have been shown to be associated with recurrence risk, other molecular markers are under investigation and should also be incorporated in clinical scenarios in the future of decision making involving adjuvant chemotherapy in localized GC including pT3N0M0 [7]. The aim of this study was to retrospectively analyze the prognostic factors of GC patients with pT3N0M0 and to clarify which ones could benefit from adjuvant chemotherapy.

## 2. Materials and Methods

We retrospectively reviewed the medical records of pT3N0M0 patients who underwent radical gastrectomy (R0 resection and number of lymph nodes retrieved > 15) between 1994 and 2020 at Chang Gung Memorial Hospital in Linkou, Taiwan. Patients with surgical mortality (within 30 days after surgery; *n* = 4), positive resection margins (*n* = 1), unknown tumor size (*n* = 1) and unknown vascular invasion status (*n* = 4) were excluded from this analysis. Stump cancer was defined as a cancer that arises in the remnant stomach after gastrectomy for benign diseases. Patients undergoing total or partial gastrectomy were recruited based on tumor size, tumor location and resection margin status. Frozen-section examination for the resection margins was performed intraoperatively by pathologists. No patient received neoadjuvant chemotherapy or postoperative irradiation therapy. Patients did not receive adjuvant chemotherapy because of a poor Eastern Cooperative Oncology Group score (≥3), comorbidities, advanced age, concerns of chemotherapy-related side effects, and patients’ will. Adjuvant chemotherapy with fluoropyrimidine-based regimens including TS-1 (40–60 mg bid for 28 days followed by 14 days of rest or 14 days followed by 7 days of rest), uracil-tegafur (UFT; 267 mg/m^2^ bid or tid) or fluorouracil (500 mg/m^2^) plus leucovorin (200 mg/m^2^) were administered to patients within 6–8 weeks after surgery. The postoperative follow-up included physical examination, esophagogastroduodenoscopy, laboratory tests (hemogram, biochemistry and tumor markers) and imaging studies, such as computed tomography or abdominal sonography. The eighth edition of the American Joint Committee on Cancer staging system was used for pathological tumor staging [8].

### 2.1. Clinical Information

Data on clinicopathological parameters, including age, sex, Charlson comorbidity index (CCI) score, tumor size, tumor location, stump cancer, type of gastrectomy (total or partial), number of lymph nodes retrieved, histological type, lymphatic invasion, vascular invasion, perineural invasion, surgical complications and adjuvant chemotherapy regimen were extracted from our institutional database.

### 2.2. Outcomes

Disease-free survival (DFS) was defined as the time from surgery to disease recurrence. Cancer-specific survival (CSS) was defined as the interval between the date of surgery and the date of death from GC. The median follow-up time was 61.01 months. The last follow-up date was 30 June 2022.

### 2.3. Statistical Analysis

Continuous variables were compared using the Mann–Whitney U-test and expressed as medians with ranges, and categorical variables were compared using Pearson’s chi-squared test or Fisher’s exact test as appropriate. We used the survminer R package to perform Kaplan–Meier survival analysis and visualization, and the ggsurvplot function to draw DFS and CSS curves with the number-at-risk table. The differences in survival distributions among the groups were compared using the log-rank test, which was performed with the survdiff function. Potentially relevant factors acquired from our univariate analysis (*p* < 0.1) were included in the multivariate analysis; both analyses were performed using Cox proportional hazard models. Patients with in-hospital mortality were excluded from the survival analysis. To establish an optimal cutoff point of tumor size for predicting recurrence, we performed a recursive partitioning analysis, which is a statistical methodology used to create a survival analysis tree [9]. All analyses were conducted using Statistical Package for the Social Sciences software for Windows (version 20.0; IBM Corp., Armonk, NY, USA) and R software (R Core Team (2021), R: A language and environment for statistical computing; R Foundation for Statistical Computing, Vienna, Austria (https://www.R-project.org; accessed on 10 March 2022)). A *p*-value of <0.05 was considered statistically significant.

## 3. Results

A total of 137 patients were included in our analysis. Adjuvant chemotherapy was administered to 54 patients (39.4%). The median duration was 11.68 months (range, 3.38–16.85 months) in 26 patients treated with TS-1. Among them, dose reduction was noted in 11 patients due to side effects and poor performance status. Eighteen patients received UFT with median duration of 23.58 months (range, 6.21–81.41 months). Intravenous fluorouracil was administered to 10 patients with a median duration of 5.52 months (range, 3.09–7.82 months). Table 1 shows the clinicopathological parameters of patients with pT3N0M0 in terms of chemotherapy. There were significant differences in age (*p* = 0.003), the number of lymph nodes retrieved (*p* = 0.042) and the presence of vascular (*p* = 0.021) or perineural invasion (*p* = 0.03) between patients treated with and without chemotherapy. No differences were observed in sex, CCI score, stump cancer, type of gastrectomy, tumor location, tumor size, histology, lymphatic invasion nor surgical complications between the chemotherapy and non-chemotherapy groups. Table 2 shows the clinicopathological features of patients with pT3N0M0 in terms of recurrence. Stump cancer (*p* = 0.046), larger tumor size (*p* = 0.016) and the presence of perineural invasion (*p* = 0.049) were found to be associated with tumor recurrence. Differences in age, sex, CCI score, type of gastrectomy, tumor location, the number of lymph nodes retrieved, tumor differentiation, the presence of vascular or lymphatic invasion and surgical complications were not evident between patients with and without recurrence. As shown in Table 3 and Table 4, stump cancer, tumor size > 4.3 cm and surgical complications were prognostic factors for DFS and CSS in our univariate analysis. After multivariate analysis, tumor size > 4.3 cm was found to be an independent unfavorable predictor of DFS (hazard ratio, 4.55; confidence interval, 1.59–12.99; *p* = 0.005) and CSS (hazard ratio, 3.97; confidence interval, 1.38–11.43; *p* = 0.011).

There were no significant differences in DFS (*p* = 0.222) and CSS (*p* = 0.126) between the chemotherapy and non-chemotherapy groups (Figure 1A,B). The rates of 3-, 5- and 10-year DFS were 84.2%, 79.3% and 79.3%, respectively, in the non-chemotherapy group, and 94.1%, 91.7% and 84.1% in the chemotherapy group. The rates of 3-, 5- and 10-year CSS were 86.7%, 80.0% and 78.0%, respectively, in the non-chemotherapy group, and 94.0%, 94.0% and 85.8% in the chemotherapy group. The survival outcomes were comparable among patients treated with varied chemotherapy regimens (UFT, TS-1 or fluorouracil) and without chemotherapy (*p* = 0.448 (Figure 2A); *p* = 0.345 (Figure 2B)). Figure 3A,B depict significantly worse rates of 5-year DFS (71.6% vs. 94%) and CSS (71.8% vs. 95.6%) of patients with tumor size > 4.3 cm than those of patients with tumor size ≤ 4.3 cm (*p* < 0.001 and *p* = 0.001, respectively). As shown in Figure 4A,B, the G3 group (tumor size > 4.3 cm and no chemotherapy) had significantly lower rates of 5-year DFS (63.6% vs. 92.3% vs. 96.3%) and CSS (60.5% vs. 95.1% vs. 96.2%) than the G1 (tumor size ≤ 4.3 cm and no chemotherapy) or G2 (tumor size ≤ 4.3 cm and chemotherapy) group. The G4 group (tumor size > 4.3 cm and chemotherapy) had favorable survival compared with the G3 group, although it was not statistically significance. A survival difference was not evident between the G1 and G2 groups.

## 4. Discussion

This study investigated the prognostic factors and the role of adjuvant chemotherapy in 137 patients with pT3N0M0 GC who underwent radical surgery with and without adjuvant chemotherapy. Twenty-one patients (15.3%) had recurrence. The risk factors of recurrence were stump cancer, tumor size and the presence of perineural invasion. The multivariate analysis showed that tumor size > 4.3 cm was an independent factor associated with DFS and CCS. The chemotherapy group was younger, had had more lymph nodes retrieved, and had higher percentages of vascular or perineural invasion than the non-chemotherapy group. The survival outcomes did not differ between the two groups.

In the era of precision or individualized medicine, tailored treatment strategies for GC patients are preferred, according to disease severity and clinicopathological characteristics, patients’ will, individuals’ general performance and comorbidities, as well as the potential benefits and disadvantages of the use of adjuvant treatment. Other advanced or modern molecular biomarkers have been increasingly adopted to select the right patients for targeted therapy or immunotherapy [7,10,11]. Nonetheless, few patients are indicated for or can afford these treatments, due to low rates of HER-2 positivity or financial toxicity [12]. Prognostic factors of DFS or CSS are easily identified and can thus be referenced for decision making for adjuvant therapy with the analysis of available clinicopathological parameters. Our study found that stump cancer, tumor size and the presence of perineural invasion were associated with higher percentages of recurrence. However, only tumor size was an independent prognostic factor for DFS and CSS. Similar to our findings, Lee et al. also identified that tumor size > 5 cm in T3N0M0 (odds ratio, 1.929; *p* = 0.030) was a risk factor for DFS [5]. Furthermore, Lu et al. reported that tumor size information can improve the accuracy of 7th edition TNM staging in predicting the survival in GC patients after R0 resection [13]. Aoyama et al. showed that tumor size was the most important prognostic factor in multivariate analysis for survival in Stage II/III patients undergoing radical gastrectomy followed by adjuvant TS-1 chemotherapy [14]. The possible explanation for these findings is that large serosal tumors may have a trend of micro-metastasis via lymphatic channels, peritoneal seeding from serosal invasion that was missed in pathological examination, or infiltrative tumor growth patterns [15,16], which are evidenced by peritoneal seeding as the predominant site of recurrence in T3N0M0 [5,17]. Our study indicated that when the factors of tumor size and chemotherapy were considered in survival analysis, adjuvant chemotherapy did not have beneficial effects for patients with small tumors, and G3 patients had the worst outcomes among our patient groups (Figure 4). Interestingly, although the G4 group had favorable survival compared with the G3 group, the survival difference was not significant. A further study recruiting large samples may further clarify the effects of adjuvant chemotherapy on patients with large tumor size.

Previous studies showed that adjuvant chemotherapy with TS-1, an oral fluoropyrimidine, improved DFS and overall survival in patients with Stage II and III GC after radical D2 surgery [18,19]. Furthermore, one trial indicated that their adjuvant eight-course-treated group had better three-year DFS than the four-course group in Stage II GC and suggested that postoperative TS-1 treatment for one year should remain as the standard of care for those patients [20]. Nonetheless, it remains unclear whether the subgroup of patients with T3N0M0 would benefit from fluoropyrimidine-based chemotherapy, since these studies did not recruit patients with pT3N0M0 (Japanese guidelines do not recommend adjuvant chemotherapy for those patients) [18,19,20,21]. However, the guidelines of National Comprehensive Cancer Network/Europe and substantial evidence suggest that Stage II and III patients should undergo adjuvant chemotherapy, chemoradiotherapy or perioperative chemotherapy [7,11,22,23]. Our results demonstrate that chemotherapy did not confer survival benefit to pT3N0M0 patients. Similar to our findings, Lee et al. found no survival differences between chemotherapy and non-chemotherapy patients with pT3N0M0 [5]. In contrast, Huang et al. indicated that using pathological features to construct a nomogram can be used to identify Stage IIA GC (90% pT3N0M0 and 10% pT1N2M0) patients who would benefit from adjuvant chemotherapy [4]. Moreover, they also found that overall survival was better in their chemotherapy group than in the non-chemotherapy group, although the chemotherapy group had higher percentages of lymph dissection number < 15 and tumor size > 2 cm than the non-chemotherapy group [4]. Furthermore, studies showed that micro-metastasis of lymph nodes were identified using cytokeratin immunohistochemical staining in node-negative (hematoxylin–eosin staining) GC, and worse survival was noted in this patient group due to stage migration, since there was no use of adjuvant therapy [24]. Therefore, we speculate that micro-metastasis to lymph nodes may have also existed in our chemotherapy group, which had higher percentages of the presence of vascular or perineural invasion than the non-chemotherapy group. This may, at least in part, explain why there were no differences in the survival outcomes between the chemotherapy and non-chemotherapy groups in this study. 

The prognostic roles of perineural invasion have been extensively explored in many solid tumors, including GC [25,26,27]. Our previous research showed that perineural invasion is an independent risk factor for distant metastasis in advanced node-negative GC [28]. Furthermore, Chen et al. identified that perineural invasion was a poor prognostic factor for Sage II-III GC after radical surgery [29]. This study showed that although patients with perineural invasion had higher rates of recurrence, perineural invasion was not a prognostic factor for DFS and CSS in univariate and multivariate analyses, implying that the prognostic role of perineural invasion in pT3N0M0 tumors is not as significant as it is for other advanced GC tumor types.

Our previous research showed that fluorouracil-based adjuvant chemotherapy did not improve overall survival in Stage II-IV GC patients with deficient mismatch repair (dMMR) or high microsatellite instability (MSI) [30]. The high-MSI GC cell lines demonstrated increased resistance to 5- fluorouracil, which may have been due to increased autophagy, since the inhibition of autophagy abolished the chemoresistance. Immune checkpoint inhibitors have been approved by Food and Drug Administration to treat high-MSI/dMMR tumors, including GC [31,32]. More efforts should be made to clarify the benefits of immunotherapy in early-stage GC, such as pT3N0M0 tumors with high-MSI/dMMR.

Recently, Tie et al. addressed the role of adjuvant chemotherapy in pT3 or pT4N0M0 colon cancer by determining circulating tumor DNA (ctDNA), indicating that the three-year recurrence-free survival was 86.4% among ctDNA-positive patients who underwent adjuvant chemotherapy and 92.5% among ctDNA-negative patients who did not [33]. Noninferiority in recurrence rates was observed in patients who underwent ctDNA-guided management compared with standard management (patients routinely received adjuvant chemotherapy after surgery). Their results suggest that a ctDNA-guided treatment strategy in pT3 or pT4N0M0 colon cancer decreased the use of adjuvant chemotherapy without compromising patient outcomes. Whether the concept is applicable to GC patients with pT3N0M0 deserves further investigation in the future.

Three fluoropyrimidine-based cytotoxic agents, including two oral forms, UFT and TS-1, and one intravenous form, 5-fluorouracil, were used in our patients. There were no differences in DFS and CSS among patients treated with the three drugs. Studies indicated that longer three-year DFS was identified in the TS-1 group than in UFT (hazard ratio = 0.81; *p* = 0.005) in T4a/b GC after radical surgery [34]. In contrast, another report showed that adjuvant UFT or TS-1 monotherapy had comparable 5-year overall survival rates (78.3% vs. 73.1%; *p* = 0.48) in GC patients with Stage II-IIIB cancer [35].

There are several key limitations to this study. First, this was a retrospective study with an inherent selection bias that was unavoidable. Second, we did not routinely examine MSI status and HER-2 positivity during the study period. Third, patients received varied chemotherapy regimens and shorter schedules of chemotherapy than originally planned in some patients, which may have affected their survival. Fourth, only a small number of cases were included. Although our results did not show significant benefits of adjuvant chemotherapy in pT3N0M0 GC, our findings provide insights for further research into adjuvant treatment in the future for this subgroup of patients.

## 5. Conclusions

Tumor size independently affected DFS and CSS in pT3N0M0 patients undergoing radical surgery with and without adjuvant chemotherapy. Although our results suggest that adjuvant chemotherapy did not provide survival benefits, the chemotherapy group had more unfavorable features than the non-chemotherapy group. A large-scale randomized trial is needed to fully clarify the role of adjuvant chemotherapy. Tailored treatment strategies should also be adopted based on tumor characteristics and patient factors, and more biomarkers, such as MSI status, HER-2 positivity or ctDNA, may provide additional useful information for treating pT3N0M0 GC.

## Figures and Tables

**Figure 1 jpm-13-00553-f001:**
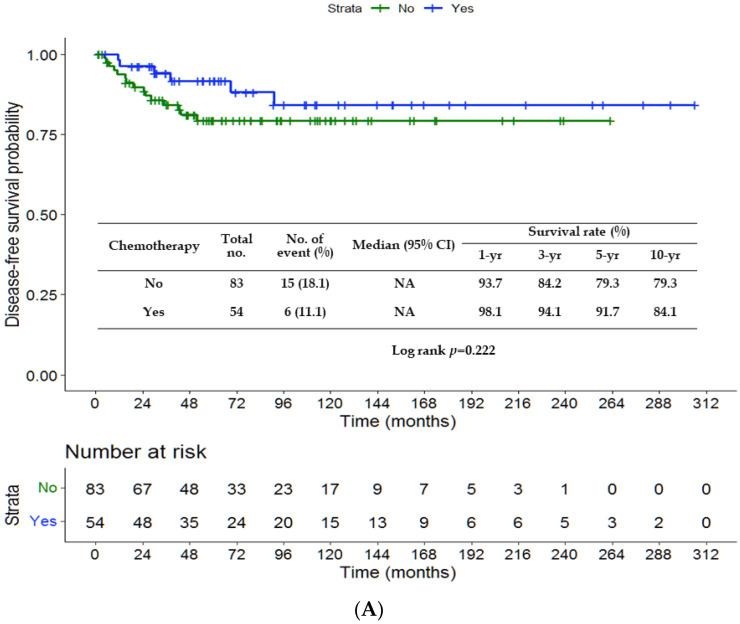
Kaplan–Meier curves of disease-free survival (**A**) and cancer-specific survival (**B**) in terms of chemotherapy. NA, not available.

**Figure 2 jpm-13-00553-f002:**
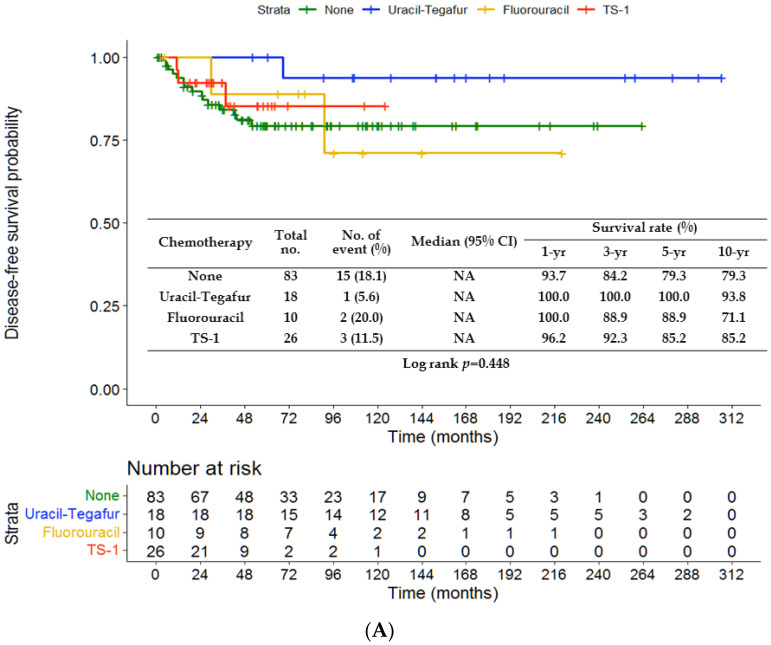
Kaplan–Meier curves of disease-free survival (**A**) and cancer-specific survival (**B**) in the non-chemotherapy group, and uracil-tegafur (UFT), fluorouracil or TS-1-treated group. NA, not available.

**Figure 3 jpm-13-00553-f003:**
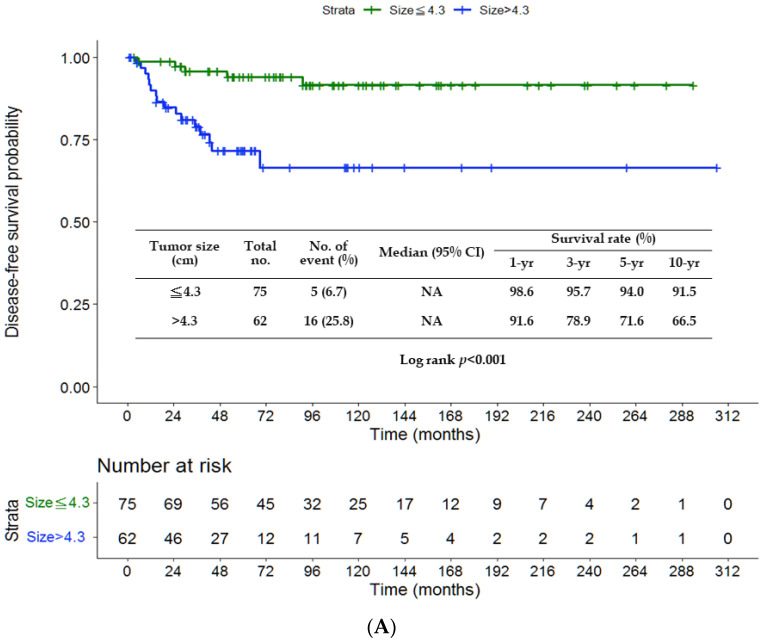
Kaplan–Meier curves of disease-free survival (**A**) and cancer-specific survival (**B**) in terms of tumor size. NA, not available.

**Figure 4 jpm-13-00553-f004:**
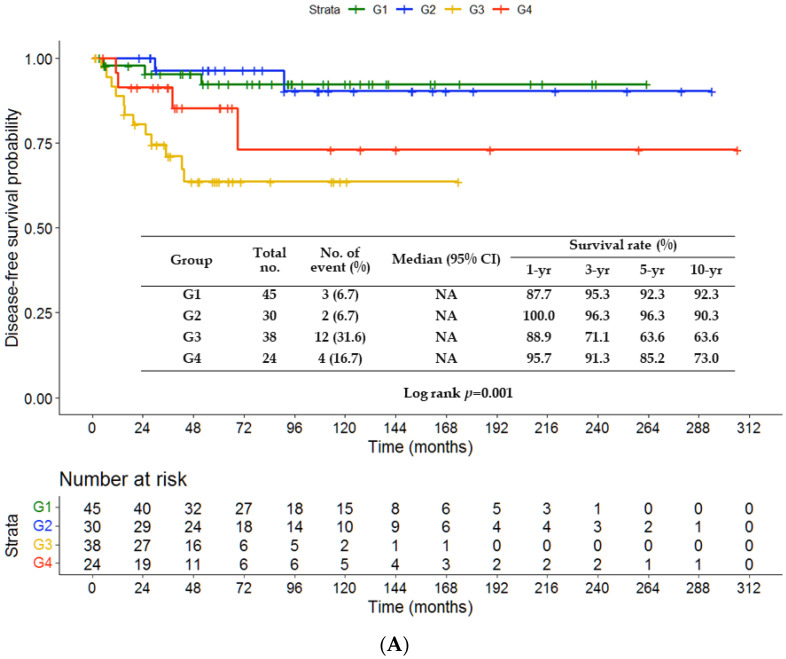
Kaplan–Meier curves of disease-free survival (**A**) and cancer-specific survival (**B**) in terms of tumor size and chemotherapy. G1, size ≤ 4.3 cm without chemotherapy; G2, size ≤ 4.3 cm with chemotherapy; G3, size > 4.3 cm without chemotherapy; G4, size > 4.3 cm with chemotherapy. NA, not available.

**Table 1 jpm-13-00553-t001:** Clinicopathological parameters of pathological T3N0M0 patients in terms of chemotherapy.

Parameter	Chemotherapy		*p*-Value
	No	Yes	
No. of cases (%)	83 (60.6)	54 (39.4)	
Age, median (range) (years)	71 (32–91)	61 (33–86)	0.003
<65	31 (37.3)	32 (59.3)	0.012
≥65	52 (62.7)	22 (40.7)	
Sex, *n* (%)			0.535
Male	58 (69.9)	35 (64.8)	
Female	25 (30.1)	19 (35.2)	
Charlson comorbidity index score, *n* (%)			0.133
0	30 (36.1)	24 (44.4)	
1	27 (32.5)	19 (35.2)	
2	12 (14.5)	9 (16.7)	
≥3	14 (16.9)	2 (3.7)	
Stump cancer, *n* (%)	6 (7.2)	0	0.081
Type of resection, *n* (%)			0.533
Total gastrectomy	24 (28.9)	13 (24.1)	
Partial gastrectomy	59 (71.1)	41 (75.9)	
Location, *n* (%)			0.405
Upper	14 (16.9)	13 (24.1)	
Middle	24 (28.9)	11 (20.3)	
Lower	45 (54.2)	30 (55.6)	
Tumor size (cm), median (range)	3.8 (1–16)	4.0 (1–11)	0.290
Number of lymph nodes retrieved, median (range)	35 (17–91)	41.5 (16–121)	0.042
Histology, *n* (%)			0.113
Differentiated	39 (47.0)	18 (33.3)	
Undifferentiated	44 (53.0)	36 (66.7)	
Vascular invasion, *n* (%)	4 (4.8)	9 (16.7)	0.021
Lymphatic invasion, *n* (%)	6 (7.2)	8 (14.8)	0.152
Perineural invasion, *n* (%)	29 (34.9)	29 (53.7)	0.030
Complications, *n* (%)	12 (14.5)	12 (22.2)	0.243

**Table 2 jpm-13-00553-t002:** Clinicopathological parameters of pathological T3N0M0 patients in terms of recurrence.

Parameter	Recurrence		*p*-Value
	No	Yes	
No. of cases (%)	116 (84.7)	21 (15.3)	
Age, median (range) (years)	68 (33–91)	62 (32–84)	0.141
<65	51 (44.0)	12 (57.1)	0.265
≥65	65 (56.0)	9 (42.9)	
Sex, *n* (%)			0.524
Male	80 (69.0)	13 (61.9)	
Female	36 (31.0)	8 (38.1)	
Charlson comorbidity index score, *n* (%)			0.274
0	46 (39.7)	8 (38.2)	
1	42 (36.2)	4 (19.0)	
2	16 (13.8)	5 (23.8)	
≥3	12 (10.3)	4 (19.0)	
Stump cancer, *n* (%)	3 (2.6)	3 (14.3)	0.046
Type of resection, *n* (%)			0.075
Total gastrectomy	28 (24.1)	9 (42.9)	
Partial gastrectomy	88 (75.9)	12 (57.1)	
Location, *n* (%)			0.108
Upper	25 (21.6)	2 (9.5)	
Middle	26 (22.4)	9 (42.9)	
Lower	65 (56.0)	10 (47.6)	
Tumor size (cm), median (range)	3.5 (1–16)	5.6 (3–10)	0.016
Number of lymph nodes retrieved, median (range)	36 (16–121)	35 (18–84)	0.185
Histology, *n* (%)			0.403
Differentiated	50 (43.1)	7 (33.3)	
Undifferentiated	66 (56.9)	14 (66.7)	
Vascular invasion, *n* (%)	11 (9.5)	2 (9.5)	1
Lymphatic invasion, *n* (%)	12 (10.3)	2 (9.5)	>0.999
Perineural invasion, *n* (%)	45 (38.8)	13 (61.9)	0.049
Complications, *n* (%)	18 (15.5)	6 (28.6)	0.207
Chemotherapy, *n* (%)	48 (41.4)	6 (28.6)	0.269

**Table 3 jpm-13-00553-t003:** Univariate and multivariate analyses of prognostic factors for disease-free survival.

Parameter	Univariate Analysis			Multivariate Analysis		
	Hazard Ratio	95% CI	*p*-Value	Hazard Ratio	95% CI	*p*-Value
Age			0.401	-		
<65 (*n* = 63)	1.45	0.61–3.44				
≥65 (*n* = 74)	1					
Sex				-		
Male (*n* = 93)	1					
Female (*n* = 44)	1.30	0.54–3.13	0.563			
Charlson comorbidity index score				-		
0 (*n* = 54)	1					
1 (*n* = 46)	0.68	0.20–2.25	0.526			
2 (*n* = 21)	1.96	0.64–6.01	0.237			
≥3 (*n* = 16)	2.39	0.71–7.97	0.158			
Stump cancer						
No (*n* = 131)	1			1		
Yes (*n* = 6)	6.02	1.75–20.76	0.004	2.61	0.69–9.85	0.158
Type of resection				-		
Total gastrectomy (*n* = 37)	1.96	0.83–4.66	0.126			
Partial gastrectomy (*n* = 100)	1					
Location						
Upper (*n* = 27)	1			1		
Middle (*n* = 35)	3.70	0.80–17.11	0.095	2.09	0.40–10.94	0.384
Lower (*n* = 75)	2.03	0.44–9.28	0.361	1.59	0.34–7.32	0.555
Tumor size (cm)						
≤4.3 (*n* = 75)	1			1		
>4.3 (*n* = 62)	5.20	1.89–14.33	0.001	4.55	1.59–12.99	0.005
Histology				-		
Differentiated (*n* = 57)	1					
Undifferentiated (*n* = 80)	1.53	0.62–3.78	0.363			
Vascular invasion				-		
No (*n* = 124)	1					
Yes (*n* = 13)	0.92	0.21–3.94	0.909			
Lymphatic invasion				-		
No (*n* = 123)	1					
Yes (*n* = 14)	1.02	0.24–4.39	0.978			
Perineural invasion						
No (*n* = 79)	1			1		
Yes (*n* = 58)	2.36	0.98–5.71	0.056	2.02	0.76–5.36	0.159
Complications						
No (*n* = 113)	1			1		
Yes (*n* = 24)	2.64	1.01–6.90	0.047	1.18	0.41–3.44	0.757
Chemotherapy				-		
No (*n* = 83)	1.79	0.69–4.61	0.228			
Yes (*n* = 54)	1					

CI, confidence interval.

**Table 4 jpm-13-00553-t004:** Univariate and multivariate analyses of prognostic factors for cancer-specific survival.

Parameter	Univariate Analysis			Multivariate Analysis		
	Hazard Ratio	95% CI	*p*-Value	Hazard Ratio	95% CI	*p*-Value
Age			0.537	-		
<65 (*n* = 63)	1.32	0.55–3.19				
≥65 (*n* = 74)	1					
Sex				-		
Male (*n* = 93)	1					
Female (*n* = 44)	1.38	0.56–3.37	0.483			
Charlson comorbidity index score				-		
0 (*n* = 54)	1					
1 (*n* = 46)	0.51	0.14–1.93	0.321			
2 (*n* = 21)	2.00	0.65–6.12	0.226			
≥3 (*n* = 16)	2.43	0.73–8.17	0.150			
Stump cancer						
No (*n* = 131)	1			1		
Yes (*n* = 6)	6.08	1.74–21.26	0.005	2.61	0.68–9.98	0.162
Type of resection				-		
Total gastrectomy (*n* = 37)	1.71	0.70–4.17	0.242			
Partial gastrectomy (*n* = 100)	1					
Location						
Upper (*n* = 27)	1			1		
Middle (*n* = 35)	7.30	0.93–57.66	0.059	3.90	0.45–33.92	0.218
Lower (*n* = 75)	4.11	0.53–32.20	0.178	3.19	0.40–25.16	0.271
Tumor size (cm)						
≤4.3 (*n* = 75)	1			1		
>4.3 (*n* = 62)	4.90	1.77–13.62	0.002	3.97	1.38–11.43	0.011
Histology				-		
Differentiated (*n* = 57)	1					
Undifferentiated (*n* = 80)	1.79	0.69–4.67	0.233			
Vascular invasion				-		
No (*n* = 124)	1					
Yes (*n* = 13)	0.99	0.23–4.27	0.988			
Lymphatic invasion				-		
No (*n* = 123)	1					
Yes (*n* = 14)	1.17	0.27–5.07	0.835			
Perineural invasion						
No (*n* = 79)	1			1		
Yes (*n* = 58)	2.22	0.91–5.44	0.081	1.87	0.69–5.01	0.216
Complications						
No (*n* = 113)	1			1		
Yes (*n* = 24)	3.11	1.17–8.26	0.023	1.39	0.47–4.11	0.551
Chemotherapy				-		
No (*n* = 83)	2.16	0.79–5.95	0.136			
Yes (*n* = 54)	1					

CI, confidence interval.

## Data Availability

No new data were created or analyzed in this study. Data sharing is not applicable to this article.

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
