# Peer review of "Prognostic Factors and the Role of Adjuvant Chemotherapy in Pathological Node-Negative T3 Gastric Cancer"

_jpm, 2023, doi:10.3390/jpm13030553_

Round 1

Reviewer 1 Report

 This is a carefully conducted retrospective analysis of prognostic factors and the impact of adjuvant chemotherapy in pT3N0M0R0 gastric cancer patients.  I have the following comments:

In lines 282 to 288 you correctly state several limitations to your study. In face of these limitations some statements throughout the manuscript should be alleviated, e.g. "our results demonstrated that chemotherapy…" (lines 225 ff) should better be rephrased to "… suggests that in the cohort studied…

In line with this suggestion, it should be clarified that the results are limited to this particular cohort and treatment regimens. 

According to current guidelines, all patients with T3 gastric cancers should be offered pre- and postoperative (neoadjuvant) chemotherapy. This important limitation should be included in the discussion and the corresponding literature be cited.

The data presented in the tables are extensive and should be shortened. The figures are informative and include most of the relevant data.

Author Response

Comments

  1. In lines 282 to 288 you correctly state several limitations to your study. In face of these limitations some statements throughout the manuscript should be alleviated, eg. "our results demonstrated that chemotherapy…" (lines 225 ff) should better be rephrased to "… suggests that in the cohort studied…
  2. In line with this suggestion, it should be clarified that the results are limited to this particular cohort and treatment regimens. 

Responses: We thank the reviewer very much for the valuable comments regarding our work. I have removed some statements about the limitation in the discussion and rephrased the sentences in the revised manuscript. We also added one sentence to clarify our results and to emphasize that further research is needed to improve patient outcomes.

The sentence “This may be explained in-part by the small volume of tested cases (Figure 4)” was removed from the end of paragraph 2 of Discussion.

The statement “; however, higher percentages of the presence of vascular or perineural invasion were identified in the chemotherapy group compared to the non-chemotherapy group “ was deleted from the paragraph 3 of Discussion.  

The sentence “It should be noted that the enrolled cases in each group were too small to draw definitive conclusions.” was also deleted from the paragraph 7 of Discussion.

  1. According to current guidelines, all patients with T3 gastric cancers should be offered pre- and postoperative (neoadjuvant) chemotherapy. This important limitation should be included in the discussion and the corresponding literature be cited.

Responses: We thank greatly for the reviewer’s remarks and we have added the statement and relevant references indicating that T3N0M0 (stage IIA) gastric cancer patients should receive adjuvant or peri-operative treatments in the revied manuscript. However, the Japanese guidelines do not recommend chemotherapy for patients with pT3N0M0. Our present study is to clarify the role of adjuvant chemotherapy in the subgroup patients with pT3N0M0.

References:

  1. Mohamed, A.A.; Gordon, A.; Cartwright, E.; Cunninghan, D. Optimising multimodality treatment of resectable oesophago-gastric adenocarcinoma. Cancers 2022, 14, 586. https://doi.org/10.3390/cancers14030586.
  2. Sato, Y.; Okamoto, K.; Kida, Y.; Mitsui, Y.; Kawano, Y.; Sogabe, M.; Miyamoto, H.; Takayama, T. Overview of chemotherapy for gastric cancer. J. Clin. Med. 2023, 12, 1336. https://doi.org/10.3390/jcm12041336.
  3. Japanese Gastric Cancer Association. Japanese gastric cancer treatment guidelines 2010 (ver. 3). Gastric Cancer 2011,14, 113-23.
  4. Ajani, J.A.; D’Amico, T.A.; Bentrem, D.J.; Chao, J.; Cooke, D,; Corvera, C.; Das, P.; Enzinger, P.C.; Enzler, T.; Fanta, P.; et al. NCCN Clinical practice guidelines in oncology. Gastric cancer, version 2.2022. J. Natl. Compr. Canc. Netw. 2022, 20, 167-192. https://doi.org/10.6004/jnccn.2002.0008.
  5. Waddell, T.; Verheij, M.; Allum, W.; Cunningham, D.; Cervantes, A.; Arnold, D.; European Society for Medical Oncology (ESMO); European Society of Surgical Oncology (ESSO); European Society of Radiotherapy and Oncology (ESTRO). Gastric cancer: ESMO-ESSO-ESTRO clinical practice guidelines for diagnosis, treatment and follow-up. Ann. Oncol. 2013, 24 Suppl 6, vi57-63.

  1. The data presented in the tables are extensive and should be shortened. The figures are informative and include most of the relevant data.

Responses: I have shortened the tables in the revised manuscript.

Reviewer 2 Report

Dear authors,

I read with interest "Prognostic Factors and the Role of Adjuvant Chemotherapy in Pathological Node-negative T3 Gastric Cancer".

This text is discussing a study that looked at 137 patients with stage T3 gastric cancer, which had not spread to the lymph nodes. It analyzed various clinicopathological factors and adjuvant chemotherapy regimens in order to determine prognostic factors for disease-free survival (DFS) and cancer-specific survival (CSS). The results showed that tumor size was an independent prognostic factor for both DFS and CSS, suggesting that larger tumors may be more likely to recur or cause death. Additionally, it found no significant difference in outcomes between those who received additional chemotherapy after surgery compared to those who did not receive any additional treatment.

Table 4 is a summary of the results from the univariate and multivariate analyses that were conducted to determine prognostic factors for disease-free survival (DFS) and cancer-specific survival (CSS). It includes information on various clinicopathological factors, such as tumor size, lymph node retrieval rate, vascular invasion percentage, perineural invasion percentage. Additionally it provides hazard ratios for each factor in relation to DFS or CSS.

Figure 1 is a graphical representation of the Kaplan-Meier curves for disease-free survival (DFS) and cancer-specific survival (CSS). It shows the differences in outcomes between patients who received adjuvant chemotherapy after surgery compared to those who did not receive any additional treatment.

I suggest:

(1) Give the formulas for Kaplan-Meier curves are based on.

(2) I do not know what the little crosses mean. In the vertical sense can be error associated with the probabilities, but in the horizontal direction? And they are just put there like in Word? Please, do theses curves in Python. Compute the errors well. Give the formulas for the computation of the error bars.

(3) The quality of the PNG is very low. Please increase the quality of the plots.

Other than that I like the atmosphere of the paper,

with best regards,
